# Optimization of Irrigation and Leaching Depths Considering the Cost of Water Using WASH_1D/2D Models

**Haruyuki Fujimaki [1],\* , Hassan M. Abd El Baki [1], Seyed Mohamad Mahdavi [2] and Hamed Ebrahimian [3]**

1   Arid Land Research Center, Tottori University, Tottori 681-0061, Japan; Hassan.wat2@yahoo.com
2   Department of Soil Science, University of Tabriz, Tabriz 51666-16471, Iran; Mohamad.Mahdavi64@gmail.com
3   Department of Irrigation and Reclamation Engineering, University of Tehran, Karaj 31587-77871, Iran; ebrahimian@ut.ac.ir
*   Correspondence: fujimaki@tottori-u.ac.jp

**Abstract:** Optimization of water use with consideration of salinity control is a crucial task for crop production. A new scheme, "optimized irrigation", was recently presented to determine irrigation depth using WASH_1D/2D which are numerical simulation models of water flow and solute transport in soils and crop growth. In the scheme, irrigation depth is determined such that net income is maximized considering the price of water and weather forecasts. To evaluate whether the optimized irrigation is also able to restrict salinity stress and avoid salinization without any intentional leaching, we carried out a numerical experiment for winter wheat grown in northern Sudan under the following scenarios: (1) Available water in the root zone is refilled using freshwater (0.17 g/L of NaCl) at every five days; (2) available water in the root zone is refilled using saline water (1.7 g/L) at every five days; (3) optimized irrigation using fresh water at 7-days interval; (4) optimized irrigation on a weekly basis using saline water; and (5) same as scenario 2, except for leaching is carried out at the middle of the growing season and leaching depth is optimized such that net income is maximized. The results showed that the optimized irrigation scheme automatically instructs additional water required for leaching at each irrigation event and maximizes the net income even under saline conditions.

**Keywords:** salinity stress; drought; solute transport; salinity control; net income

## 1. Introduction

Irrigation-induced salinity is a major threat for the sustainability of agriculture in arid and semi-arid regions. Many studies have reported how salinity reduces yield [1,2]. According to Food and Agriculture Organization of the United Nations (FAO), salinity is already causing yield reduction in 37 million ha, nearly one-third of upland-irrigated farmlands, all over the world [3,4]. Qadir et al. [5] estimated that the global annual cost of salt-induced land degradation in irrigated areas is 27.3 billion USD. Climate change may worsen this issue in many irrigated areas where rainfall is predicted to decrease. Phogat et al. [6] reported that, according to future climate projections, irrigation schedules without a significant leaching fraction might lead to a high salt build-up in the soil. In addition, as more water in drainage canals is getting used for irrigation owing to lack of fresh water, secondary salinity will be more serious than ever. To control salinity in the root zone for better crop growth, more water than that required to meet crop evapotranspiration must be applied to leach excessive soluble salts out [7]. Such an intentional "over-irrigation" is called leaching, which is the primary measure and widely practiced as the most effective method for removing salts from the rootzone. In order to keep or

even enhance farmers' incomes, while securing sustainability of productivity of lands, it is essential to determine the appropriate amount of leaching (leaching depth) that keeps the salt contents of the root zone within an acceptable range for crop production, but this may compromise water savings.

To determine leaching depth, the procedure of FAO irrigation and drainage paper 29 [7] has been commonly used. Although the FAO procedure has an advantage of simplicity for being applied in all regions of the world, the equation presented in the guideline is based on unrealistic steady-state solution of solute transport. In addition, table of predicted yield reduction and soil salinity level for major crops is given, but the evidence is not traceable and therefore reliability of the data is questionable. Relying upon the conventional scheme may have led over-use of precious water and/or reduced yield and net income. Letey and Feng [8] stated there is a considerable doubt in the application of the conventional steady-state scheme for estimating leaching requirement. They pointed out that this scheme results in an erroneous estimation of leaching requirement due to failure to account for transient conditions. We may improve the efficiency of leaching and net income by fully utilizing the fruits of basic studies for predicting the movement of water, heat, and solute in soils. Corwin et al. [9] applied a one-dimensional numerical model, UNSATCHEM, to evaluate how much the conventional steady-state model overestimates leaching requirement. Hanson et al. [10] applied a two-dimensional model, HYDRUS_2D, to analyze the leaching process. Ramos el al. [11] evaluated the accuracy of a one-dimensional model, HYDRUS-1D, for simulating soil salinity under an irrigated cropland. Isidoro and Grattan [12] developed a model to predict root zone salinity under different irrigation practices and soil types. In general, their model [12] predicted lower electrical conductivity of the root zone than the FAO steady-state scheme in the growing season. Hence, considering the site-specific features could lead to lower leaching requirements. Qureshi et al. [13] applied SWAP model [14] to seek optimum groundwater table depth and irrigation schedules for controlling soil salinity in Iraq. Corwin and Grattan [15] compared various steady-state and transient schemes for calculating leaching requirement. They reported that conventional steady-state schemes overestimate leaching requirement which increase demand on limited freshwater resources. Moreover, overestimation of leaching requirement will result in a greater volume of drainage water and therefore more leaching of salts and agrochemicals (e.g., fertilizers and herbicides). The biggest reason for assuming steady-state conditions might be simply because of the lack of sufficient accessible computer processing power to solve the complex solute transport equations under transient conditions using numerical analysis techniques at that time it was first presented in 1976. As a consequence, they recommended developing a user-friendly transient and sophisticated transient model for calculating leaching requirement during a crop growing season.

Still, such numerical models have never been used to dynamically determine leaching depth at each time during a crop growing season. Leaching depth can be determined such that simulated net income, considering the cost of water, is maximized under given water price. A similar scheme has been developed for determining irrigation depth by Fujimaki et al. [16] using WASH_1D/2D, numerical simulation models of water flow in soil and crop growth using weather forecast (e.g., rainfall events, solar radiation, wind velocity, relative humidity) (or past weather data). Abd El Baki et al. [17–19] evaluated the effectiveness of the proposed scheme through field experiments in a sandy field using potato, sweet-potato, and soybean. The new scheme may be regarded as a climate-smart irrigation, since the scheme can automatically respond to changing climate. Since the numerical models can also consider salinity stress in addition to drought stress, a question has arisen as to whether the new scheme, optimized irrigation, is able to restrict salinity stress in addition to drought stress and avoid severe salinization in the root zone without any planned leaching. Short-term optimization may not necessarily give optimum results in long term. Therefore, one of the purposes of this study is to evaluate whether optimized irrigation is also able to control salinity automatically. If optimized irrigation can control salinity, it means that leaching is carried out at each irrigation event simultaneously. In general, leaching is carried out as a deliberate practice and only once or twice at each cropping season may be effective, because if drainage amount is the same, the higher the salinity in the root zone is, the larger

salts are removed. Ayers and Westcot [7] recommended holding leaching until salinity in the root zone exceeds the tolerance level of the crop. Thus, we will present a new scheme to determine leaching depth such that simulated net income is maximized under given water price. Another purpose of this study is to evaluate the new scheme to determine leaching depth (optimized leaching) in which leaching is performed only once during a cropping season can attain higher net income than the optimized irrigation scheme. To avoid large experimental errors and deviations which often mask key findings and focus on theoretical benefit, we carried out a numerical experiment rather than field ones in this study. We admit that numerical experiments are not fully immune to errors such as mass balance, but results are unique (i.e., zero standard deviation) and repeatable. Therefore, we believe that the first experiment to evaluate a new scheme should be numerical.

## 2. Methods

### 2.1. Virtual Net Income

We assume that a farmer can obtain virtual net income, $I_n$ ($ $ \text{ha}^{-1}$), at each irrigation although a farmer can receive income after harvest in reality. $I_n$ is calculated on the assumption that yield of sold part of crops is proportional to cumulative transpiration at each irrigation interval [16–19]:

$$I_n = P_c \varepsilon \tau_i k_i - P_w W - C_{ot} \tag{1}$$

where $P_c$ is the producer's price of crop ($ $ \text{kg}^{-1}$ DM), $\varepsilon$ is transpiration productivity of the crop (produced dry matter (kg ha$^{-1}$) divided by cumulative transpiration (kg ha$^{-1}$, 1 mm = 10,000 kg ha$^{-1}$)), $\tau_i$ is cumulative transpiration between two irrigation events (1 mm = 10,000 kg ha$^{-1}$), $k_i$ is the income correction factor, used to avoid underestimation of $I_n$ due to smaller transpiration rate in the initial growth stage; $P_w$ is the price of water ($ $ \text{kg}^{-1}$), $W$ is the irrigation depth (1 mm = 10,000 kg ha$^{-1}$), and $C_{ot}$ is other costs (e.g., fertilizers and pesticides) ($ $ \text{ha}^{-1}$). The above equation may be modified to consider the difference in the sensitivity to stress among growing stages and the effect of stresses on price of crop. Since we focused on optimized leaching depth and salinity stress in this study, we applied the simplest form of net income equation. To maximize $I_n$, we need a model of the response of $\tau_i$ to $W$. In this study, we used the WASH_1D model which is freely available at http://www.alrc.tottori-u.ac.jp/fujimaki/download/WASH_1D/ with source code and case files used in this study [16]. WASH_1D solves governing equations (parabolic partial differential equations) for one-dimensional movement of water, solute and heat, including Richards' equation of water flow and convection and dispersion equation (CDE) of heat and salt movements in soils with the finite difference method.

### 2.2. Numerical Model of Root Water Uptake and Crop Growth

Transpiration rate, $T_r$ (cm s$^{-1}$), is calculated by integrating the water uptake rate, $S$ (s$^{-1}$), over the plant root zone:

$$T_r = \int_0^{drt} S dz \tag{2}$$

where $z$ is depth and $d_{rt}$ is depth of the lower boundary of root zone. In the macroscopic root water uptake model, $S$ is given as the product of potential transpiration, $T_p$ (cm s$^{-1}$), reduction coefficient which implies both drought (water) and salinity stress, $\alpha_w$, and normalized root density distribution, β:

$$S = T_p \beta \alpha_w \tag{3}$$

the $T_p$ is assumed to be proportional to reference evapotranspiration, $ET_o$, calculated with the Penman-Monteith equation [20] and basal crop coefficient, $k_{cb}$

$$T_p = ET_o k_{cb} \tag{4}$$

the $k_{cb}$ is expressed as a function of cumulative transpiration as:

$$k_{cb} = a_{kc}[1 - \exp(b_{kc}\tau)] + c_{kc} - d_{kc}\tau^{e_{kc}} \tag{5}$$

where $a_{kc}$, $b_{kc}$, $c_{kc}$, $d_{kc}$, and $e_{kc}$ are fitting parameters. The crop coefficient ($k_{cb}$) varies from zero to around 1.3 depending on the percentage of land covered by crop in each growth stage. By using cumulative transpiration as the independent variable instead of the conventional day after sowing, factors such as nutrient deficit or salinity stress affecting on plants growth and development can be reflected. The income correction factor $k_i$ in Equation (1) is given as the ratio of mean $k_{cb}$ to $k_{cb}$ at each day:

$$k_i = \frac{\overline{k_{cb}}}{k_{cb}} = \frac{\int k_{cb}d\tau}{\tau_f k_{cb}} = \frac{(a_{kc} + c_{kc})\tau_f - \frac{a_{kc}}{b_{kc}}\left[\exp\left(b_{kc}\tau_f\right) - 1\right] - \frac{d_{kc}\tau_f^{e_{kc}+1}}{e_{kc}+1}}{\tau_f k_{cb}} \tag{6}$$

where $\tau_f$ is maximum cumulative transpiration for the crop. The β is described as:

$$\beta = 0.75(b_{rt} + 1)d_{rt}^{-b_{rt}-1}(d_{rt} - z + z_{r0})^{b_{rt}} \tag{7}$$

where $b_{rt}$ is a fitting parameter and $z_{r0}$ is the depth below which roots exist (cm). When $b_{rt} = 1$, the β linearly decreases with depth below $z_{r0}$. The $d_{rt}$ is also expressed as a function of τ as:

$$d_{rt} = a_{drt}[1 - \exp(b_{drt}\tau)] + c_{drt} \tag{8}$$

where $a_{drt}$, $b_{drt}$, and $c_{drt}$ are fitting parameters. Using $k_{cb}$ and $d_{rt}$ parameters as functions of τ instead of days after sowing, the plant growth may respond more dynamically to drought or salinity stresses. The $\alpha_w$ is a function of matric potential, ψ (cm) and osmotic potential, $\psi_o$ (cm):

$$\alpha_w = \frac{1}{1 + \left(\frac{\psi}{\psi_{50}} + \frac{\psi_o}{\psi_{o50}}\right)^p} \tag{9}$$

where $\psi_{50}$, $\psi_{o50}$, and p are fitting parameters. Further description of the model such as the computation of water flow as well as solute and heat movement in the vadose zone is available in Fujimaki et al. [16].

## 2.3. Optimized Irrigation

When irrigation interval is given, we may find W which gives the highest $I_n$ in Equation (1). In WASH_1D, the W is optimized using the Golden Section Method [21] with an accuracy of 3% of searched range (10 trials) between 0 and user-specified maximum value (i.e., $W_{max}$). When actual climatic data is available or actual irrigation depth is different from the optimum value, "update run" (step 2 in Figure 1) should be carried out to estimate current status and initial condition using actual irrigation record and the climatic data (step 1). Then, after obtaining quantitative weather forecast until the next scheduled irrigation (step 3), a simulation consisting of 10 trials is run to determine the optimum irrigation depth to be applied (step 4). The irrigation is then performed. This cycle continues until the final irrigation.

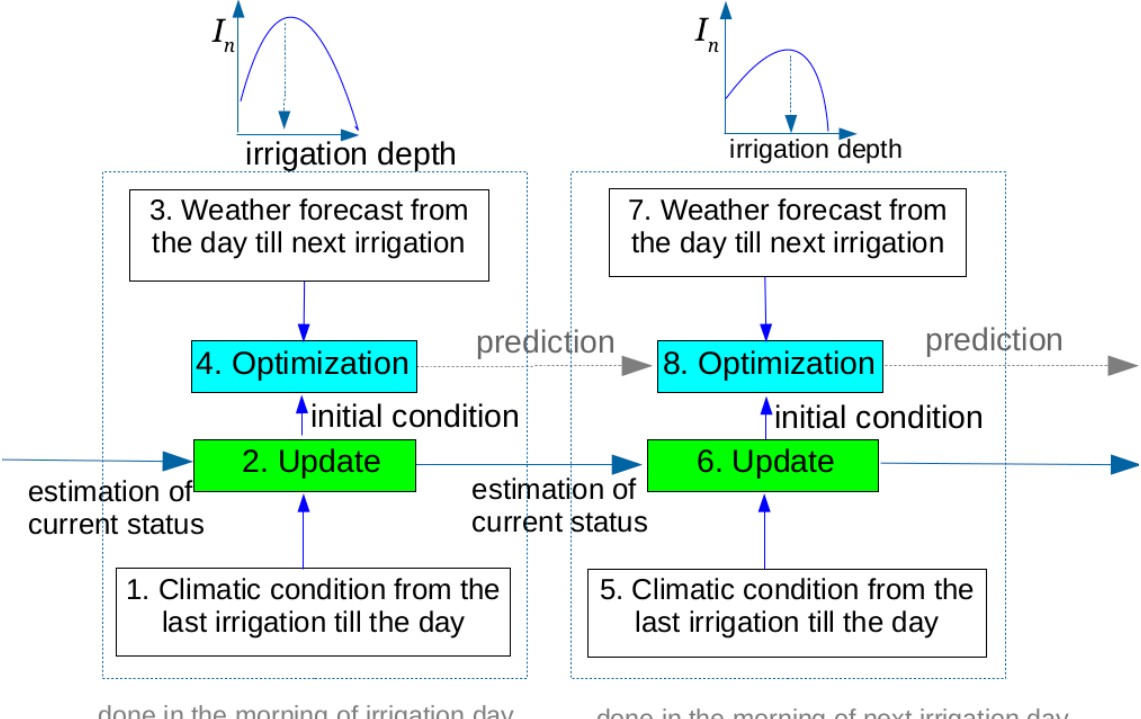

**Figure 1.** Schematic of the routine procedure of optimized irrigation (Numbers and arrows denote steps and data use, respectively).

### 2.4. Refilling Irrigation

Salinization tends to be more serious in arid regions than semi-arid or dry sub-humid climate if the salinity of irrigation water is similar. One of the advantages of the optimized irrigation may be the reduction of downward percolation loss by considering forecasted rain. In salt-affected farmlands, such downward percolation loss may wash salts from the root zone and act as unintentional leaching. Therefore, a simpler scheme is applied to determine irrigation depth in order to return volumetric water content to field capacity in the root zone. This may be common practice as instructed in Allen et al. [21]. We would call this the "refilling scheme". We may determine irrigation intervals dynamically by frequently checking if available water in critical zone still remains, but it is less practical and more laborious for farmers. Thus, we would suggest fixing the interval at a value which does not cause significant reduction in cumulative transpiration during the entire growing season (i.e., maintaining soil water content above the threshold value for drought stress). Newly added user interface for this option is shown in the screen shot (Figure 2).

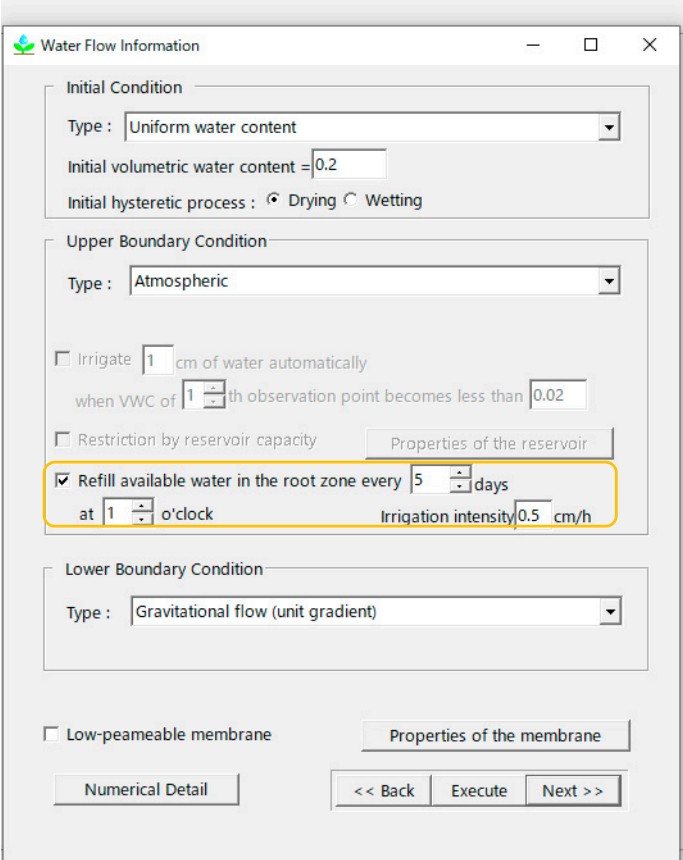

**Figure 2.** Screen shot of the user interface showing the newly added options for refilling the soil profile as surrounded by a yellow rectangle.

### 2.5. Optimized Leaching

We may apply the same procedure as optimized irrigation until the end of the growing season, but irrigation to refill available water in the root zone is performed at a fixed interval. Unlike optimized irrigation whose interval is generally shorter than a week, weather forecast for a few months is usually not reliable even if it is available. Therefore, climatic data of the last or from a representative year may be used instead. Timing of leaching may be dynamic by monitoring the salinity of critical zone, but it may also be laborious. Thus, we would suggest carrying out leaching at the middle of each growing season. In the early days in the growing season, build-up of salts may still not be so harmful while in the late days in the growing season, it may be too serious to achieve expected yield.

### 2.6. Conditions for the Numerical Experiment

Winter wheat cultivation in northern Sudan was simulated under the following scenarios:

S1. Refilling irrigation using freshwater (0.17 g/L of NaCl) at every 5 days.
S2. Refilling irrigation in the root zone is refilled using saline water (1.7 g/L of NaCl) at every 5 days.
S3. Optimized irrigation on a weekly basis using fresh water.
S4. Optimized irrigation on a weekly basis using saline water.
S5. Same as S2 except for optimized leaching is carried out at the middle of the growing season.

Northern Sudan was selected because it is a typical hyper-arid zone and experimental results are not affected by uncertainty in rainfall. Winter wheat is a common crop in the region.

In this study, one dimensional model, WASH_1D is used for simplicity and wheat is usually irrigated with entire irrigation method such as border or sprinkler irrigation. In case of drip irrigation

for larger and more sparsely cultivated crop such as tomato or cucumber, two-dimensional version, WASH_2D may be used. Soil hydraulic properties of a soil taken in Dongola, Sudan were determined with an evaporation method [22] and were used in the simulation as shown in Figure 3. Note that wafer flux and volumetric water content numerically solved using the Richards equation are simultaneously used in the convection–dispersion equation (CDE). Parameter values used for calculating solute dispersion coefficient, $D$ (cm$^2$ s$^{-1}$) used in the CDE are listed in Table 1.

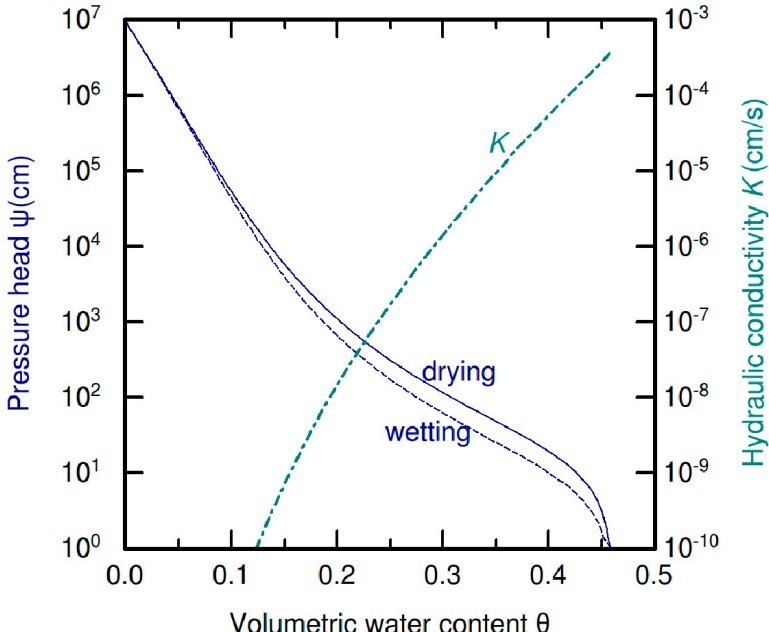

**Figure 3.** Soil water retention and hydraulic conductivity curves of the soil taken in Dongola, Sudan.

**Table 1.** Parameter values for solute and heat movement for the soil.

| Parameter | Unit | Value | Equation |
|---|---|---|---|
| $\lambda$ | cm | 0.36 | |
| $a_s$ | - | 0.26 | Equation (11) |
| $b_s$ | - | 16 | Equation (12) |
| $c_s$ | - | 3 | |
| $d_s$ | - | 0.27 | |
| $a_h$ | W cm$^{-2}$ s$^{-1}$ | 0.0078 | |
| $b_h$ | W cm$^{-2}$ s$^{-1}$ | 0.0052 | |
| $c_h$ | - | 18,041 | Equation (13) |
| $d_h$ | W cm$^{-2}$ s$^{-1}$ | 0.0011 | |
| $e_h$ | - | 5.96 | |
| $\alpha_{max}$ | - | 0.27 | |
| $\alpha_{min}$ | - | 0.18 | Equation (14) |
| $a_l$ | - | 12 | |
| $b_l$ | - | 5.2 | |

In this study, we assumed that no salts are uptaken by plant roots nor supplied through fertilizer. Salinity build-up is thus simulated sorely with the sink term of water by plant roots and evaporation from the soil.

$$D = D_m + D_i \tag{10}$$

The mechanical dispersion coefficient, $D_m$ (cm$^2$ s$^{-1}$), is proportional to pore water velocity, $V$ (cm$^2$ s$^{-1}$).

$$D_m = \lambda |V| \tag{11}$$

where $\lambda$ is dispersivity (cm). The ionic diffusion coefficient, $D_i$ (cm$^2$ s$^{-1}$) is the product of that in free water, $D_{i0}$, and tortuosity, which is computed with the following empirical function:

$$D_i = D_{i0}\{a_s\ \{1 - \exp[(-b_s\theta)^{cs}]\} + d_s\theta\}$$ (12)

where $a_s$, $b_s$, $c_s$, and $d_s$ are soil-specific empirical parameters. Those values as well as the dispersivity are taken from a soil from Mushaqar, Joran, having a soil water retention curve similar to the soil we used.

In addition, WASH_1D is able to simulate heat movement because evaporation rate, which is calculated independently from transpiration in WASH_1D, strongly depends on soil temperature. Ionic diffusion is also affected by temperature. Parameter values for the dependences of thermal conductivity, $kh$, (W cm$^{-2}$ s$^{-1}$) and albedo, $\alpha_R$, on water content and properties are also displayed in Table 1.

$$k_h = a_h + b_h\left(\frac{\theta}{\theta_{sat}}\right) - (a_h - d_h)exp\left[-c_h\left(\frac{\theta}{\theta_{sat}}\right)^{eh}\right]$$ (13)

where $\theta_{sat}$ is saturated volumetric water contents and $a_h$, $b_h$, $c_h$, $d_h$, and $e_h$ are soil-specific empirical parameters. Those parameter values were determined by curve-fitting of thermal conductivities measured with the single probe method at different water content.

$$\alpha_R = \frac{\alpha_{max} - \alpha_{min}}{1 + (a_{al}\overline{\theta_\ell})^{bal}} + \alpha_{min}$$ (14)

where $\theta_\ell$ is average volumetric water contents in the top 0.5 cm, $\alpha_{max}$, $\alpha_{min}$, $a_{al}$, and $b_{al}$ are soil specific empirical parameters. Those parameter values were determined by curve-fitting of values measured in the field at different water content.

Climatic data from 10 to 31 March 2019 and those from 1 December 2019 till 9 March 2020, were used as continuous data. Dongola is located in a hyper-arid zone and no rain was recorded during the periods. Dongola is the capital of the state of Northern in Sudan. It lies on the west bank of the Nile River. Wheat is supposed to be sown on 1 December and harvested on 31 March according to the common wheat growing season in the study area. Parameters used in the crop growth model and Equation (1) are listed in Tables 2 and 3, respectively. We determined parameter values of Equation (5), by curve fitting of data of cumulative transpiration estimated at each boundary between growing stages from the table presented by FAO [21].

**Table 2.** Parameter values of plant growth and stress response function used for the numerical simulation.

| Parameter | Unit | Value | Equation |
|---|---|---|---|
| $a_{kc}$ | - | 1.19 | |
| $b_{kc}$ | - | −0.32 | |
| $c_{kc}$ | - | 0.1 | Equation (5) |
| $d_{kc}$ | - | 0.00000112 | |
| $e_{kc}$ | - | 3.9 | |
| $b_{rt}$ | - | 1 | Equation (7) |
| $z_{r0}$ | cm | 2 | |
| $a_{drt}$ | cm | 55 | |
| $b_{drt}$ | - | −0.05 | Equation (8) |
| $c_{drt}$ | cm | 5 | |
| $\Psi_{50}$ | cm | −4000 | |
| $\Psi_{o50}$ | cm | −8000 | Equation (9) |
| $P$ | - | 3 | |

**Table 3.** Parameter values in the net income equation (Equation (1)).

| Parameter | Description | Unit | Value |
|:---:|:---:|:---:|:---:|
| $P_c$ | Price of crop | USD kg$^{-1}$ | 0.05 |
| $\varepsilon$ | Transpiration productivity | - | 0.002 |
| $P_w$ | Price of water | USD m$^{-3}$ | 0.01 |
| $C_{ot}$ | Other costs | USD ha$^{-1}$ | 50 |

Depth of lower boundary was set at 60 cm. Thickness of top and bottom elements were set at 0.75 and 8 cm, respectively. Initial conditions for water, solute, and heat were uniform: Volumetric water content at 0.2 cm$^3$/cm$^3$, concentration at 0.17 (for S1 and S3) or 8.3 g/L of NaCl (for S2, S4, and S5), and temperature at 28 °C, respectively. In this study, the initial concentration was set at nearly the same average salinity at the end of the season to evaluate whether the average salinity in the root zone is increased in S4 and S5. If the average salinity resulted in significantly larger value than that at the beginning, salts may build-up in long term and such a scheme may not be sustainable. Upper boundary condition (at the soil surface) is atmospheric plus irrigation while lower boundary conditions for water, solute and heat were zero gradient for pressure, concentration and temperature, respectively.

In reality, weather forecasts somewhat underestimate or overestimate the values even for a few days, but in this study we assumed that weather forecasts provide completely accurate forecast to avoid stresses or over-irrigation caused by inaccurate weather forecasts. When heavy rain is forecasted, optimum irrigation depth will be zero. But if rain does not occur, plants may suffer from drought stress. In contrast, if unexpected heavy rain occurs just after irrigation, the irrigation is regarded as over-irrigated. In hyper-arid climate, weather forecast tends to be accurate since most of the inaccuracy comes from rainfall prediction. Therefore, unlike actual optimized irrigation scheme, there is no difference between the results of optimization and update runs and thereby the results of last optimization runs were used as initial conditions for next optimization runs. In actual optimized irrigation scheme, the results of last update run are used as initial conditions for next optimization run.

Optimized irrigation started from 15 January and refilling scheme was carried out every five days until then to avoid any drought stress. In scenario 5, leaching was carried out on 31 January, mid-day across the growing period. The mass balance error is calculated by summing the values of all inflows and outflows and absolute errors are divided by cumulative flux. Water, solute and energy balance errors of the model were very low and negligible. For instance, these errors were 0.5, 0.0, and 0.07%, respectively, for S3.

## 3. Results and Discussion

### 3.1. Cumulative Irrigation, Transpiration, and Drainage

The cumulative irrigation, transpiration, and drainage for each scenario are shown in Figure 4. Scenarios 1 (refill and freshwater, S1) and 3 (optimized irrigation and freshwater, S3) gave almost the same cumulative transpiration. Both scenarios attained 95% of potential transpiration and yield. S3 gave smaller irrigation depth at the final stage than S1 because the refilling scheme tried to refill water until field capacity even when crop water requirement is decreased and therefore met by applying less water enough to recover field capacity throughout the root zone. Scenario 2 (refill and saline, S2) had the lowest cumulative transpiration due to a low osmotic potential. Note that cumulative irrigation of S2 was lower than S1 owing to lower transpiration. S2 had larger drainage than S1 and S3, probably owing to smaller transpiration compared to S1 and S3. Cumulative irrigation of Scenario 4 (optimized irrigation and saline water, S4) was larger by about 27% than S3, and showed almost the same cumulative transpiration as S1 and S3. Interestingly, S4 had almost the same cumulative irrigation (including leaching) as Scenario 5 (optimized leaching) in the final stage. These results indicate that optimized irrigation is also able to control salinity automatically. Scenario 5 had consistently lower transpiration than S4. Although slope (daily transpiration rate) became steeper than S4 just after

leaching, S5 could not finally catch up with S4. This result may imply that the leaching which was carried out just once at the middle of the growing season was inefficient. The highest net income occurred around 150 mm leaching, although this point is not quite distinct (Figure 5). Note that the highest net income occurred at a point which gave lower cumulative transpiration. This "highest" net income is attained under the restriction of given timing of leaching (i.e., at the middle of the growing period).

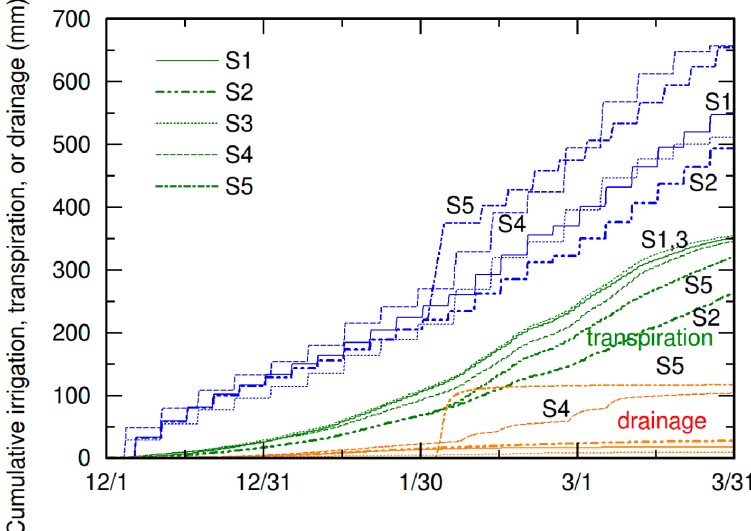

**Figure 4.** Time evolution of cumulative irrigation, transpiration and drainage for each scenario. Blue, green, and orange lines represent irrigation, transpiration, and drainage, respectively.

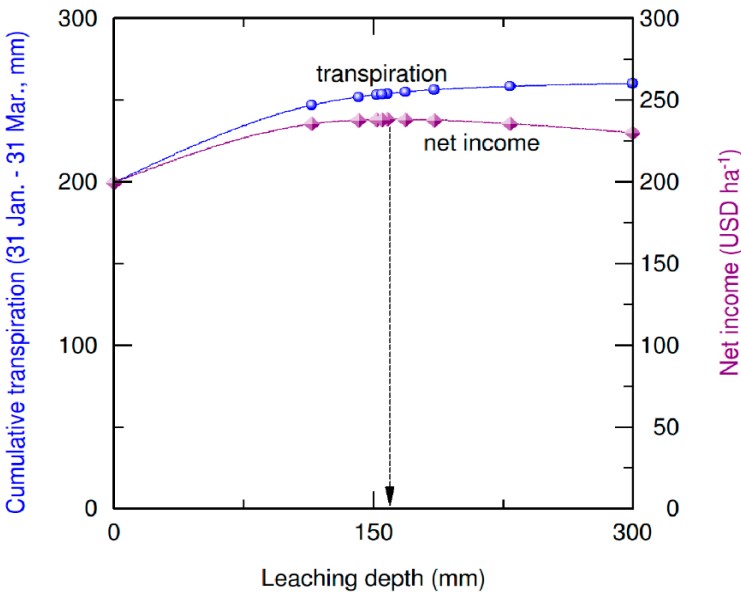

**Figure 5.** Transpiration amount and increment of net income from 31 January to 31 March as a function of leaching depth in Scenario 4. Those leaching depths are trials during the search of maximum with the golden section method.

### 3.2. Disadvantages and Advantages of the Proposed Leaching Scheme

One of the disadvantages of less frequent leaching scheme as S5 is that such large amount of water may not be available at the middle of the growing season. Another potential drawback is if drainage of the farmland is poor, it may cause lack of oxygen in the root zone, whose negative

effect is not considered in the simulation model. In addition, in clayey soils, preferential flow along macropores which reduces efficiency of leaching often occurs when a large amount of water is applied in a short time [7]. Under those situations, optimized leaching should be avoided, and optimized irrigation is recommended instead. It is true that errors in weather forecast somewhat reduce the benefit of optimized irrigation, but optimized leaching is also not immune from underestimation of drought/salinity stresses or overuse of water for leaching caused by difference between climatic data and actual climatic condition. Therefore, averaging of results using climatic data for the last three or four years would be recommended. One of the advantages of optimized irrigation (S4) is that it uses short term weather forecast and discrepancy caused by error in forecast may be corrected by update run which uses actual weather data if available. Following this line of research, Mirás-Avalos et al. [23] developed a new tool, Irrigation-Advisor, which is based on weather forecasts, for providing irrigation recommendations in vegetable crops. This approach instructed lower irrigation volumes, resulting in water savings without yield reduction.

### 3.3. Soil Salt Content

Profiles of salt content per unit volume of soil for S2, S4 and S5 are shown in Figure 6. The salts remaining in the soil down to 60 cm at the end of S2, S4 and S5 scenarios were, respectively, 166, 104, and 107 mg cm$^{-2}$, while the initial salt storage at the beginning of the simulations was 100 mg cm$^{-2}$. Each scenario had a peak near the center of the root zone. Largest accumulation occurred in S2 owing to lack of downward percolation. In this scenario, less irrigation water application (i.e., without leaching) resulted in higher salt accumulation and lower transpiration. Wang et al. [24] reported similar results for spring wheat in Northwest China. He et al. [25] also stated that inappropriate irrigation with brackish water caused soil salinization, causing salinity stress for winter wheat accordingly, in the North China Plain. Optimized leaching (S5) also had a sharp peak at the depth of 26 cm because after leaching at the middle of the growing season, no leaching was carried out. Low salt concentration below 30 cm of S5 also implies leaching was not carried out very efficiently, because root activity in the lower part of the root zone is low. Optimized irrigation (S4) has the lowest and deepest peak because unintentional leaching was carried out until the last irrigation (Figure 6). The fact that both optimized irrigation and optimized leaching did not significantly increased salt storage indicates the sustainability of both schemes.

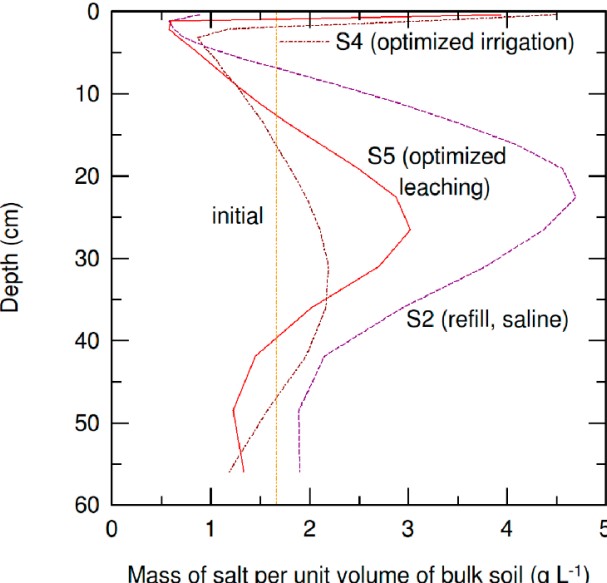

**Figure 6.** Salt content over the soil profile at the end of the growing season for scenario 2, 4, and 5.

### 3.4. Net Income

The net income comparisons among the five scenarios are shown in Figure 7. Although S3 gave smaller irrigation depth, net income of it is almost the same as S1 because cost for water is relatively low. Cultivation without any leaching (S2) apparently reduced the net income. By deliberately (S5) or unintentionally (S4) performing leaching, net income somewhat recovered to those of potential values under non-saline conditions in spite of larger cost for water. Optimized irrigation (S4) attained higher net income than optimized leaching (S5). Peragon et al. [26] reported that annual gross income could be increased by considering leaching when irrigating olive orchards with saline water in the province of Jaen (south Spain). Using a geographic information system, the authors defined areas where either water blending or leaching fraction was required. These practices could increase the annual gross income in the province.

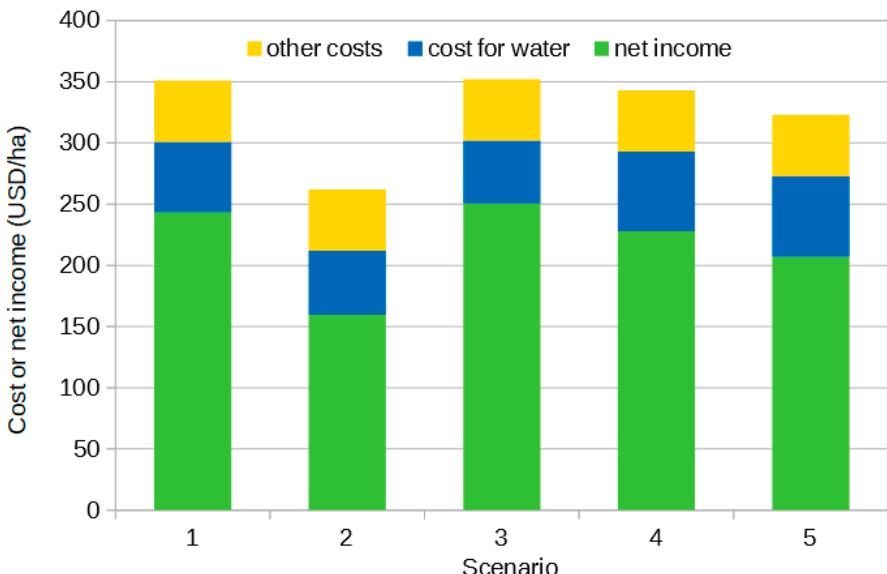

**Figure 7.** Comparison of costs and net income among the scenarios.

## 4. Conclusions

We have presented new schemes for irrigation and leaching using a numerical simulation model of water flow and solute transport in soil and crop growth, WASH_1D. We carried out a numerical experiment of winter wheat cultivation in Northern Sudan to evaluate two new schemes including optimized irrigation and optimized leaching considering the price of water and weather forecast in order to restrict salinity stress in addition to drought stress and avoid severe salinization in the root zone. Results indicated that optimized irrigation is also able to control salinity unintentionally. The scheme proposed in this study (i.e., optimized irrigation) not only allowed the use of saline water in irrigation without excessive soil salinization and crop yield reduction, but also maximized farmer's net income. Net income of optimized leaching was slightly lower than optimized irrigation; still, leaching depth can be optimized such that net income is maximized. To clarify which scheme tends to give higher net income, more numerical and field experiments under various combination of crop, soil and climate would be required. In this study, we used a detailed model based on partial differential equations to precisely predict water flow and solute movement in soils. Balance-based simpler model such as the MOPECO-Salt Model [27] or the AquaCrop [28] may also be used to optimize irrigation depth considering the cost of water. Further studies are needed for evaluating the applicability of those simpler models for optimizing leaching depth.

**Author Contributions:** Conceptualization, H.F.; methodology, H.F. and H.M.A.E.B.; software, H.F.; validation, H.F., and S.M.M.; writing—original draft preparation, H.F.; writing—review and editing, S.M.M., H.E. and H.M.A.E.B.; All authors have read and agreed to the published version of the manuscript.

**Funding:** This work was supported by JSPS KAKENHI Grant Number JP19KK0168.

**Conflicts of Interest:** The authors declare no conflict of interest.

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
