# Peer review of "Optimization of Irrigation and Leaching Depths Considering the Cost of Water Using WASH_1D/2D Models"

_water, doi:10.3390/w12092549_

Round 1

Reviewer 1 Report

The authors present a manuscript on “Optimization of leaching depth considering the cost  for water using a numerical simulation model of 3 solute transport in soils and root water uptake”. It was interesting research, the optimized irrigation has been received great concern in the arid region all the time. The paper is well-written. This manuscript was appropriate for publication in WATER, some comments for consideration.

Line 60 :” the model predicted lower electrical conductivity of the root zone than the FAO steady-state scheme in the growing season”, which model ?  Ref 9-12 ?

Figure 4, can you improve this figure, looks more clearly

Author Response

Thank you very much for reading our manuscript and giving us valuable comments.

We have revised our manuscript according to your comments a listed in the uploaded file.

We hope that our manuscript is now ready for publication.

Reviewer 2 Report

The manuscript entitled “Optimization of leaching depth considering the cost for water using a numerical simulation model of solute transport in soils and root water uptake” (Reference number Water-911012) authored by H. Fujimaki, H.M. Abd El Baki, S.M. Mahdavi and H. Ebrahimian describes the results from a numerical simulation exercise in which 5 irrigation scenarios (combining varying salt concentrations in the irrigation water and several irrigation schedulings) were tested. The method called “optimized irrigation” provided the best results when compared to the control with freshwater. However, the authors do not present a calibration of the model used. They concluded that the WASH_1D model can automatically instruct the additional water required for leaching at each irrigation event and maximize the net income even under saline conditions, so it would be useful for decision support. This manuscript fits well within the scope of Water. However, it presents severe flaws that prevent its acceptance for publication.

First of all, the authors did not perform any statistical analysis of their data, preventing from obtaining sound conclusions and must be taken into account for discussing the results.

Second, Methods need further information for describing what was exactly done. In fact, there are many unclear aspects such as the calibration and validation of the model.

Third, the results obtained are briefly described. Differences among scenarios could be not significant (authors did not perform any statistical analysis). Moreover, discussion is missing and authors must explain better and discuss further their results bearing in mind that they are outputs from simulations from a model with an unclear validation for the conditions of the studied area.

Finally, English needs revising all over the manuscript for improving wording and sentence structure.

Therefore, I recommend the rejection of this manuscript since it does not reach the high-quality standards for being published in Water.

In the following pages, I provide the authors with several comments and suggestions.

Specific comments to the authors:

Abstract:

The abstract does not include any conclusion from the study. Besides, I suggest to indicate the relevance of the study in the first sentences of the abstract; I mean, why did you perform this study? Why optimizing leaching depth is relevant? English needs improvements.

Line 12: “Optimized irrigation”, did you name the scheme that way? The comma after 2D must be removed.

Line 15: “forecasts” instead of “forecast”.

Line 16: “avoid salinization without any planned leaching”, this is unclear. Please, re-phrase it.

Line 18: “0.17 g/L” of what? Please, indicate.

Lines 19-20: I would use “on a weekly basis” instead of “whose interval is 7 days”.

Lines 23-24: How was this additional water calculated by the model? Include “event” before “and maximizes”. “conditions” instead of “condition”.

Keywords:

Please, do not use words that already appear in the title (leaching, in this case).

Introduction:

This section is brief but well structured. The references used are relevant and updated. However, English needs several improvements and the last paragraph is confusing and incomplete regarding the comments on previous research.

Line 32: Please, consider removing “because of lost crop production”.

Lines 34-37: These sentences are confusing. Please, re-phrase them.

Line 38: Include “that” before “required”.

Line 40: Include “for removing salts from the rootzone” after “effective method”.

Lines 40-43: Re-phrase to “In order to keep or even enhance farmers’ incomes, while securing the sustainability of land productivity, it is essential to determine the appropriate amount of irrigation water to be leached (leaching depth) that keep salt contents in the rootzone within an acceptable range for crop production, but this may compromise water savings”.

Line 44: The number of the reference given for the FAO method is not correct. Besides, include “the” before “procedure”.

Lines 44-97: This paragraph is very long and messy. It may confound the readers because of the English language, which needs relevant improvements (too many to go over them in detail). Besides, there are many citations to previous works but it is not clear what were the main findings of these works in relation to the current study. The relevance of the current study can be inferred but it is not clearly stated, as well as the main objective. Finally, the last sentence seems not convincing as field studies could corroborate the findings reported by the numerical simulations performed. The authors claim “To avoid experimental errors which often mask key findings and focus on theoretical benefit, we carried out a numerical experiment rather than field ones in this study”, which is misleading because numerical experiments are also subjected to errors and uncertainties. In this sense, authors need to validate their modelling scheme with actual measurements.

Methods:

There are many unclear aspects in the description of the Methods that need to be corrected, especially when describing the model used. The description of the calibration and validation processes is messy and confusing.

Line 101: Do you mean at each irrigation event? Why this assumption? Is it not unreal? How can you calibrate/validate a model that provides economic returns for a farmer after each irrigation event?

Equation 1: Does this equation come from the literature?

Lines 106-107: How is this correction factor estimated? Does it vary depending on the crop species?

Line 110: “stages” instead of “stage”.

Line 122: How did you calculate the rootzone?

Line 125: “implies” instead of “implicates”. What do you mean by “normalized root density distribution”? Normalized to what?

Line 128: Remove “the proportionality coefficient is called”. The citation should be number 21 instead of number 4.

Line 130: How did you estimate those “fitting” parameters?

Line 137: “may respond more dynamically to drought or salinity” instead of “may be more dynamically responded to drought or salinity”.

Lines 145-147: These sentences do not make sense. Please, re-phrase them.

Lines 149-150: Remove “using actual irrigation record and climatic data”.

Lines 152-153: How does the model know which one is the final irrigation event?

Figure 1: This scheme is unclear. A workflow should be used to reflect the routine for the calculations. Besides, the caption should be self-explanatory and stand alone. Please, re-phrase it.

Line 158: This is not true. Salinization severity does not depend on climate only.

Line 160: “forecasted” instead of “forecast”. Besides, this would depend on the accuracy of the prediction of rainfall amounts, which happens to be low, difficult and unreliable. In any case, the method employed for forecasting rainfall must be mentioned.

Lines 164-168: This is confusing because of English mistakes. Please, check and re-phrase.

Line 191: Re-phrase to “Screen shot of the user interface showing the newly added options for refilling the soil profile surrounded by a yellow rectangle”.

Lines 196-197: This is also unreliable and subject to a large degree of uncertainty as rainfall is highly variable from year to year.

Lines 198-199: Why do you suggest this and not in any other date? Explanations need to be provided, as the ones you stated are not convincing.

Line 201: Include “the” before “following”. Why did you select winter wheat in Sudan for your simulations? A reason must be given.

Line 203: Include “of NaCl” after “1.7 g L-1”. By the way, why do you use “g L-1” here when you used “g/L” in the abstract? Please, be consistent with the units.

Lines 204-205: “on a weekly basis” instead of “whose interval is 7 days”.

Line 206: “S2” instead of “2”.

Lines 214-216: Unclear and confusing. Please, re-phrase.

Line 218: Remove “is” before “mechanical”.

Line 220: “the product of that”, what do you mean by “that”?

Line 221: “and tortuosity, which is computed with the following empirical equation” instead of “and tortuosity and the tortuosity is described with an empirical equation”.

Table 1: The title needs re-phrasing to be self-explanatory. Use “Equation” instead of “Remark”. In the column for units, what is W as a unit? Watt? Indicate if these values are used in the simulations carried out in the current study and how these parameters have been obtained (literature, calibration, etc). Moreover, there is a mistake as “amin” appear twice in the table with two different values.

Figure 3: The caption is not correct as the figure does not depict all the existing hydraulical properties of a given soil. Please, re-phrase it so it describes what is actually shown in the figure.

Equation 13: There are mistakes with the symbols used. You used “a” instead of alphas. Please, check and correct the equation.

Lines 239-240: Why using data from 10-31 March 2019 instead of for 2020? In fact, this is not “continuous data”. Include “an” before “hyperarid”.

Line 242: Why supposed?

Table 2: The title needs re-phrasing to be self-explanatory. Use “Equation” instead of “Remark”. Indicate how these parameters have been obtained (literature, calibration, etc).

Table 3: There is an error in the units for other costs. Please, correct.

Line 250: According to the simulation scenarios formerly described, concentration should be 1.7 g L-1 instead of 8.3 g L-1 as stated here. Please, correct where needed.

Lines 251-253: This is unclear. Please, re-phrase it.

Lines 254-260: First, this paragraph is confusing and needs English revision and corrections. Second, rainfall forecast in arid zones is also subject to uncertainties and inaccuracies. The explanations given are not satisfactory.

Line 261: Remove “at”.

Lines 263-264: How were these errors calculated? Errors with respect to what? Which was the reference? Why are they only presented for S3?

No statistical analyses have been performed and the analysis of the outputs from the model is not clear because it is not described.

Results and Discussion:

The results obtained are briefly described. Differences among scenarios could be not significant but authors did not perform any statistical analysis. English needs revising all over the section for improving wording and sentence structure. Moreover, discussion is missing and authors must made an effort to explain better and discuss further their results bearing in mind that they are outputs from simulations from a model with an unclear validation (if any, since the authors did not present any validation or calibration of their model) for the conditions of the studied area.

Line 269: Include “the” before “same”.

Line 270-271: This sentence can be reduced: “Both scenarios attained 95% of potential transpiration and yield”.

Line 274: “a low” instead of “decreased”.

Lines 275-276: This sentence is confusing. Please, re-phrase it.

Line 278: Include “showed” before “almost”.

Figure 4: Improve this figure. The lines are very difficult to distinguish and the numbers do not help. Besides, you should use the same nomenclature as in the text so, instead of 1, 2, 3, 4 and 5, you must use S1, S2, S3, S4 and S5. Re-phrase the caption, which is not self-explanatory.

Line 280: Not sure about this conclusion.

Lines 283-286: Unclear. Please, re-phrase.

Figure 5: This figure needs further explanations. Where do the points come from? They are more than the scenarios simulated in the current study. Which one refers to which scenario? The caption needs re-phrasing. In fact, readers would not know from where this figure come from and how data have been obtained.

Lines 290-295: OK, but what do you recommend to users when one of these situations happens? How can the model be helpful under these situations?

Lines 295-298: This sentence is messy and potentially confusing. Please, re-phrase it.

Line 301: Remove “it is”.

Line 302: Re-phrase to “Soil salt content status”.

Line 303: This is not true. Figure 6 does not display data for S1 and S3.

Line 303-305: Re-phrase to “The salt concentrations remaining in the soil down to 60 cm at the end of S2, S4 and S5 scenarios were, respectively, 166, 104 and 107 g cm-2, while the soil salt concentration at the beginning of the simulations was 100 g cm-2”.

Line 306: Remove “loss”.

Lines 309-310: English needs corrections in this sentence. Please, re-phrase it.

Figure 6: The caption should be more explanatory. Please, use different colours for the lines representing each scenario.

Line 321: “low” instead of “small”.

Line 322: Remove “of them”.

Lines 325-326: This statement needs further explanations.

Line 328: Remove one of two “in”.

Line 330: “Low salt concentration” instead of “low concentration”.

Line 332: Make reference to Figure 6 in this sentence.

Line 334: I would remove “than initial condition”.

Figure 7: The caption should be self-explanatory and describe better the contents of the figure.

Conclusions:

The conclusions are quite long when compared to the absence of Discussion. In addition, they are not backed up by any statistical analysis, so they are not sound.

Line 337: These irrigation schemes are not so new.

Line 338: “a” instead of “an”.

Lines 338-343: This sentence is too long and messy. Please, reduce it.

Line 346: “but also maximized farmer’s net income”, depend on which scenario you use as reference.

Line 347: Remove “which are very crucial from sustainable point of view”.

Lines 348-349: How? This has not been discussed in the previous section of the manuscript.

Lines 350-355: These sentences are not conclusions from your study and must be removed.

References:

References must be edited according to journal guidelines, please check because I detected some mistakes, for instance:

Lines 383-385: This citation does not follow the same editing format as the former ones: The initials of authors are not separated by the next author using a semi-colon (;), the title of the journal is not abbreviated.

Line 422: This DOI does not correspond to the reference provided.

Some references that authors may find useful for improving their manuscript:

Abdel-Gawad, G.; Arslan, A.; Gaihbe, A.; Kadouri, F. The effects of saline irrigation water management and salt tolerant tomato varieties on sustainable production of tomato in Syria (1999-2002). Agric. Water Manag. 2005, 78, 39-53.

Alkhasha, A.; Al-Omran, A. Simulated tomato yield, soil moisture, and salinity using fresh and saline water: experimental and modeling study using the SALTMED model. Irrig. Sci. 2019, 37, 637-655.

Feng, G.; Zhang, Z.; Zhang, Z. Evaluating the sustainable use of saline water irrigation on soil water-salt content and grain yield under subsurface drainage condition. Sustainability 2019, 11(22), 6431, doi: 10.3390/su11226431

Malash, N.; Flowers, J.T.; Ragab, R. Effect of irrigation systems and water management practices using saline and non-saline water on tomato production. Agric. Water Manag. 2005, 78, 25-38.

Mirás-Avalos, J.M.; Rubio-Asensio, J.S.; Ramírez-Cuesta, J.M.; Maestre-Valero, J.F.; Intrigliolo, D.S. Irrigation-Advisor – A decision support system for irrigation of vegetable crops. Water 2019, 11, 2245, doi: 10.3390/w11112245

Nachshon, U. Cropland soil salinization and associated hydrology: Trends, processes and examples. Water 2018, 10(8), 1030, doi: 10.3390/w10081030

Author Response

(The authors gave the same response as above.)

Reviewer 3 Report

Dear Editor,

Paper entitled “Optimization of leaching depth considering the cost for water using a numerical simulation model of solute transport in soils and root water uptake” is a well written paper with studied topic well in the Journal scope, presenting interesting approach with a possibility of application in agricultural production.

Optimization of water use and efficiency in crop production is an important topic and considering all, I would suggest acceptance for publishing after some small revision.

Title: Something like “Optimization of irrigation and leaching depth considering the cost for water using a WASH_1D/2D numerical simulation model” – or similar, would maybe be more accurate

L 93-95: It seems like some words are missing here, please check, perhaps it should be something like: “Another purpose of this study is to evaluate the new scheme to determine leaching depth AND optimized leaching, performed only once during a cropping season, AND IF IT can attain higher net income than the optimized irrigation scheme.”

L95-97: Here, maybe rephrase this sentence because although I understand what you wanted to say, it is also true that numerical models should be validated with field data to gain wide usage and here it seems that experimental data are not necessary when in the end, they are the most relevant ones…

L195-196 “weather forecast for a few months is usually not reliable even if it is available. Therefore, climatic data of last or representative year may be used instead”

and L254-256: “In reality, weather forecasts give somewhat different values as actual ones, but it is assumed that weather forecast gives completely accurate forecast in this study to avoid stressed or over-irrigation caused by inaccurate weather forecast. In arid climate, weather forecast tends to be accurate”

These sentences seem to be contradictory – I get the point and context, but still maybe rephrase

Good luck!

Author Response

(The authors gave the same response as above.)

Round 2

Reviewer 2 Report

The revised version of the manuscript entitled “Optimization of irrigation and leaching depths considering the cost of water using WASH_1D/2D models” (Reference number Water-911012-v2) authored by H. Fujimaki, H.M. Abd El Baki, S.M. Mahdavi and H. Ebrahimian represents a clear improvement from the original submission. Authors have taken all my comments into account and provided satisfactory responses to my concerns. I appreciate the effort made by the authors and congratulate them on their work.

Although the authors did not perform any statistical analysis of their data, they replied that numerical experiments have zero variance and should be employed for designing new practices. This is correct, but in this sense the results are preliminary..

I still feel that authors could make an additional effort for discuss further their results their results further.

Finally, English still needs revising all over the manuscript for improving wording and sentence structure.

Therefore, I recommend a minor revision of this manuscript since it does not reach the high-quality standards for being published in Water.

Specific comments to the authors:

Abstract:

Line 20: “3) Optimized irrigation using fresh water at 7-day intervals” instead of “3) Optimized irrigation whose interval is 7 days using fresh water”.

Introduction:

I still feel that English needs improvements.

Lines 33-35: Re-phrase to “Phogat et al. [6] reported that, according to future climate projections, irrigation schedules without a significant leaching fraction might lead to a high salt build-up in the soil”.

Line 37: Include “that” after “than”. The problem here is that you continued the sentence after “required”, otherwise, you could omit “that”.

Line 41: Include “the” before “appropriate”.

Line 82: “changing” instead of “changed”.

Line 92: “exceeds the tolerance” instead of “exceeds to tolerance”.

Line 94: Remove “is” prior to “performed”.

Line 96: Remove “caused by”.

Line 98: “to” instead of “from”.

Line 100: “numerical” instead of “numerically”.

Methods:

Line 129: Remove “proportionality coefficient is called”.

Line 133: “of the conventional day after” instead of “of conventionally used day after”.

Line 145: You can include the name of the author of reference [16].

Line 160: “regions” instead of “land”.

Line 161: “similar” instead of “the same level”.

Line 169: Include “the” before “entire”.

Line 191: “of the last or from a representative year” instead of “of last or representative year”.

Line 195: “obtain” or “achieve” instead of “gain”.

Lines 213-215: Please, re-phrase this sentence, which is unclear.

Line 222: “similar soil water retention curve”, similar to what?

Lines 228-229: This sentence needs re-phrasing: “In addition, WASH_1D is able to simulate heat flow because evaporation rate, which is calculated independently from transpiration, strongly depends on soil temperature”.

Line 236: “contents” instead of “content”.

Line 240: “contents” instead of “content”.

Line 247: What do you mean by “border”?

Lines 256-257: This sentence is confusing. Please, re-phrase it.

Line 260-262: This sentence does not make sense and is repetitive. In fact, the second part of the sentence does not explain why weather forecasts can avoid stresses caused by inaccurate weather forecasts. This needs to be re-phrased.

Line 263: “since most of the inaccuracy comes from rainfall prediction” instead of “since mostly erroneous climatic data is the amount of rain”.

Lines 264-266: Again, this can confuse the readers. Please, re-phrase it.

Line 271: “negligible” instead of “ignorable”.

Results and Discussion:

Lines 279-280: This sentence is confusing. Please, re-phrase it.

Line 281: “due to” instead of “owing to”.

Lines 288-289: This sentence does not make sense. Please, re-phrase it.

Lines 290-291: Please, re-phrase to “The highest net income occurred around 150 mm leaching, although this point is not quite distinct (Figure 5)”.

Figure 5: What do you mean by “golden section search”? It is unclear.

Line 300: “clayey soils” instead of “clayey soil”.

Line 301: “when a large amount of water is applied over a short time”.

Line 306: “last” does not make sense. Either use “the last” or “at least”.

Line 309: Include “In this sense” or “In this context” or “Following this line of research” before “Mirás-Avalos”.

Line 321: There is a typo: “ininappropriate”. Please, correct to “inappropriate” or “unsuitable”.

Line 322: “causing” instead of “conducted”.

Figure 6: You can re-phrase the caption to “Salt content over the soil profile at the end of the growing season for scenarios 2,4 and 5”.

Lines 339-341: Re-phrase to: “Using a geographic information system, the authors defined areas where either water blending or leaching fraction was required. These practices could increase the annual gross income in the province”.

References:

Line 407: A problem seems to have occurred with this reference. The number 16 has no article associated.

Line 413: Here, the reference has no number assigned.

Author Response

Thank you very much for your precious time.

We have edited our manuscript as listed in the uploaded file.

We hope our manuscript is now acceptable.
